# A phase I clinical trial to evaluate the tolerability and safety of an allogeneic iPSC-derived iNKT cell and α-GalCer-pulsed autologous DC combination therapy for patients with recurrent and advanced head and neck cancer: A study protocol

Tomoya Kurokawa[1,2], Tomohisa Iinuma[2], Haruna Ebisu[1], Tomoha Yanagidaira[1], Yosuke Inaba[1], Tadami Fujiwara[1], Yoko Hattori[1], Tominaga Fukazawa[3], Takahiro Aoki[3,4], Haruhiko Koseki[3], Hideki Hanaoka[1], Toyoyuki Hanazawa[2], Shinichiro Motohashi[4]*

1 Clinical Research Center, Chiba University Hospital, Chiba, Chiba, Japan, 2 Department of Otorhinolaryngology, Head and Neck Surgery, Graduate School of Medicine, Chiba University, Chiba, Chiba, Japan, 3 Laboratory for Developmental Genetics, RIKEN Center for Integrative Medical Sciences, Yokohama, Kanagawa, Japan, 4 Department of Medical Immunology, Graduate School of Medicine, Chiba University, Chiba, Chiba, Japan

* motohashi@faculty.chiba-u.jp

## Abstract

Natural killer T (NKT) cells show intense tumor-killing activity through direct and indirect pathways. However, humans have less than 0.01% of NKT cells in the peripheral blood, making it difficult to apply NKT cells for cancer treatment. We have successfully produced invariant NKT cells derived from induced pluripotent stem cells (iPSC-iNKT cells) and demonstrated the tolerability of this product in a previous phase 1 clinical trial. Although the iPSC-iNKT cells showed substantial anti-tumor activity against various tumor cell types when combined with α-Galactosylceramide-pulsed autologous dendritic cells (DC/Gal) in non-clinical experiments, the tolerability of this combination therapy for humans has not demonstrated yet. Thus, we planned this first-in-human phase 1 open-label clinical trial to demonstrate data on tolerability and safety, as well as explore the immunological changes that occur following this combination treatment. We hypothesize that the sequential induction of nasal submucosal DC/Gal injection and intra-arterial administration of the iPSC-iNKT cells is tolerable and has a favorable safety profile for cancer patients. This trial provides vital and fundamental information for next-phase clinical trials and future applications of iPSC-iNKT cells for various cancer patients (jRCTa030220741; URL: https://jrct.mhlw.go.jp/en-latest-detail/jRCTa030220741).

**Data availability statement:** No datasets were generated or analyzed during the current study. De-identified research data will be made publicly available upon completion and publication of the study.

**Funding:** HK: 21bk0104119h0001, 24bk0104173h0001 Japan Agency for Medical Research and Development https://www.amed.go.jp/en/index.html The funder didn't play any role in the study design, data collection and analysis, decision to publish, nor preparation of the manuscript.

**Competing interests:** Regarding COIs, H. Koseki and S. Motohashi received grants from BrightPath Biotherapeutics Co., Ltd. All other authors declare that they have no relevant conflicts of interest. This does not alter our adherence to PLOS ONE policies on sharing data and materials.

**Abbreviations:** ALT, alanine aminotransferase; AST, aspartic aminotransferase; CTCAE, common terminology criteria for adverse events; CTL, Cytotoxic T lymphocyte; DC, dendritic cells; DLT, dose limiting toxicity; EDC, electronic data capture; ECOG, Eastern Cooperative Oncology Group; GCP; good clinical practice; HLA, human leukocyte antigen; IDMC, Independent Data Monitoring Committee; mMRC, modified Medical Research Council; MTD, maximum tolerated dose; NYHA, New York Heart Association; PS, performance status; RECIST, response evaluation criteria in solid tumors; α-GalCer, α-Galactosylceramide.

## Introduction

Head and neck cancer is the sixth most common cancer worldwide. Although more than 90% of head and neck cancer cases are squamous cell carcinomas, other related pathologies can occur [1]. For non-advanced head and neck cancer, treatment methods such as oral surgery and intensity-modulated radiotherapy are associated with a high success rate while maintaining quality of life for the patient. The primary approaches for advanced head and neck cancer are concurrent chemoradiotherapy and salvage surgery, as needed, or radical resection surgery followed by postoperative chemoradiotherapy [1]. However, despite these highly invasive treatments, the 5-year survival rate for many advanced head and neck squamous cell carcinoma cases remains around 40% to 50% [2–5]. Furthermore, new therapeutic strategies using targeted agents and immune therapy, such as cetuximab, nivolumab, and pembrolizumab, have been developed in recent years [6,7]. Although they demonstrated a response rate of 15% to 35% in head and neck cancer patients [4,8,9], they remain limited. Therefore, additional therapeutic methods are needed for this disease.

Invariant natural killer T (iNKT) cells are a unique type of lymphocyte that express T cell antigen receptors (TCRs) and natural killer receptors (NKRs) on their cell surface [10]. Unlike other T cell types, iNKTs recognize glycolipids, such as alpha-galactosylceramide (αGalCer) [11]. When presented by antigen-presenting cells via CD1d, a monomorphic MHC-like molecule, the glycolipid induces rapid iNKT cell activation. Activated iNKT cells can then recognize CD1d-presented lipid antigens on cancer cells, exerting direct cytotoxicity against CD1d+ tumor cells regardless of their major histocompatibility complex (MHC) expression, including through release of perforin and granzymes. They can also promote broader antitumor immune responses that indirectly harm cancer cells by activating NK cells and cytotoxic T lymphocytes (CTLs) through release of interferon (IFN)-γ and other cytokines [12–14]. In clinical trials targeting non-small cell lung cancer, intravenously administering autologous activated iNKT cells or αGalCer-pulsed autologous dendritic cells (DC/Gal) showed safety and effectiveness, with no serious adverse events (AEs) observed, confirming the induction of anti-tumor immune responses derived from NKT cells [15,16]. Although the number of peripheral blood iNKT cells is generally decreased in cancer patients, a clinical trial revealed that head and neck cancer patients can retain the peripheral blood iNKT cells. Additionally, unlike regular T cells, iNKT cells also maintain durable IFN-γ production and proliferative capacity following conventional radiation therapy [17]. These promising data suggest that iNKT cell immunotherapy may be useful for head and neck cancer. In clinical studies conducted at Chiba University Hospital, we found that DC/Gal injection into the nasal concha submucosa could induce their migration to cervical lymph nodes and activate iNKT cells [18,19]. From this evidence, we conducted a phase 1 trial of DC/Gal submucosal administration for recurrent and advanced head and neck cancer patients who were not eligible for curative surgery or irradiation. No serious AEs occurred, with eight out of nine cases showing increased anti-tumor immune activity and one case showing tumor shrinkage [20].

However, iNKT cells naturally exist at a sparse frequency of 0.01% to 0.1% among white blood cells, with considerable variation among patients, suggesting limitations in their therapeutic utility. Therefore, we developed a novel therapeutic strategy involving the administration of a sufficient number of iNKT cells, which were differentiated, matured, and proliferated from induced pluripotent stem cells (iPSCs) to head and neck cancer patients via tumor feeding artery including the peripheral branch of the maxillary artery. The iPSC-derived iNKT (iPSC-iNKT) cells are validated by the expression of Va24 iNKT cell-specific rearranged TCR and IFN-γ production. In non-clinical trials, iPSC-derived iNKT (iPSC-iNKT) cells showed an *in vitro* killing effect on a variety types of cancer cell lines (S3 and S4 Files). Furthermore, the iPSC-iNKT cells showed tumor growth inhibition was observed in human head and neck cancer cells transplanted into mice, with no safety concerns found [21].

From these findings, we conducted a phase 1 trial to evaluate the tolerability and safety of super-selective arterial infusion of the iPSC-iNKT cells for patients with recurrent and advanced head and neck cancer (jRCT2033200116). The results indicated that the iPSC-iNKT cells showed tolerability at a $1.0 \times 10^8$ cells/m$^2$ dose without the combined use of DC/Gal [21]. Hence, in the current phase 1 trial, we aimed to demonstrate the tolerability of combining the iPSC-iNKT cells and DC/Gal as the next step towards implementing this therapeutic approach for head and neck cancer patients.

## Materials and methods

### Trial designs, aim, and settings

This clinical trial is a non-blinded, single arm phase 1 trial targeting patients with recurrent or advanced head and neck cancer who are resistant or intolerant to standard of care. This trial aims to investigate the tolerability, safety, pharmacokinetics, pharmacodynamics, and anti-tumor effects of DC/Gal treatment followed by the administration of iPSC-iNKT cells. This trial starts with a screening period for eligibility after the informed consent process, then a seven-day DC/Gal preparation period for culturing the DC/Gal from each participant. This is followed by a 19-day treatment period, with a single DC/Gal injection and administration of the iPSC-iNKT cell product five days after the DC/Gal injection. We avoid same-day administration of the investigational product to more than two participants, with a minimum interval of at least seven days before administering the investigational product to the following participant. This provision is to prevent multiple individuals from experiencing acute fatal adverse events on the same day. The monitoring period consists of two phases: a treatment period, during which patients are hospitalized and closely monitored for dose-limiting toxicities (DLTs), and an observation period, during which patients are followed as outpatients with three visits over 28 days (Figs 1 and 2).

The dosing regimen involves both a submucosal injection of $1.0 \times 10^8$ DC/Gal (fixed dose) and tumor arterial infusion of iPSC-iNKT cells at one of two dose levels: initial dose (first dose) cohort ($3.0 \times 10^7$ cells/m$^2$) or second dose cohort ($1.0 \times 10^8$ cells/m$^2$) administered at once. We employ the three-by-three (3 + 3) design for dose escalation in this trial. Ultimately, the maximum tolerated dose (MTD) will be determined based on a total of six participants with no or one DLT occurrences (Fig 3).

### Participants

**Inclusion criteria.** Patients who meet all the following criteria are eligible.

1) Patients with recurrent or advanced head and neck cancer, refractory or intolerant to standard of care, and who have evaluable lesions that can be treated with an intra-arterial infusion to the tumor.

2) Patients who have not been given a previous therapy within 1 month. Any type of previous treatment is acceptable.

3) Patients must be at least 20 years old, but younger than 80 years old.

4) Patients must have an Eastern Cooperative Oncology Group (ECOG) Performance Status of 2 or less.

| | | Screening | DC/Gal Prep. | Treatment Period | | | | | | Observation Period | | | Termination |
|---|---|---|---|---|---|---|---|---|---|---|---|---|---|
| Day | | −42〜−8 | -7 | 1 | 6 | 7 | 8 | 13 | 19 | 1 | 15 | 28 | — |
| Allowance (days) | | | | +3 | +3 | — | — | | ±3 | ±3 | ±7 | ±7 | +7 |
| Informed consent | ● | | | | | | | | | | | | |
| Demographics | | ● | | | | | | | | | | | |
| Eligibility | | ● | | | | | | | | | | | |
| Pregnancy test | | ● | | | | | | | | | | ● | ● |
| Infection test | | ● | | | | | | | | | | | |
| Chest X ray | | ● | | | | | | ● | | ● | ● | ● | ● |
| ECG | | ● | | | | | | | | | | | |
| Enhanced CT or MRI | | ● | | | | | | | | | ● | | ● |
| DC/Gal preparation | | | ● | | | | | | | | | | |
| DC/Gal injection | | | | ● | | | | | | | | | |
| iPS-iNKT cell administration | | | | | ● | | | | | | | | |
| CBC/Laboratory test | | ● | | ● | ● | | | ● | | ● | ● | ● | ● |
| Coagulation test | | ● | | | ● | | | | | | | | |
| Urology | | ● | | ● | ● | | | | | ● | ● | ● | ● |
| Vital sign | | ● | ● | ◄————————————————————————————————————► | | | | | | | | | |
| Body weight | | ● | | ● | ● | | | ● | | ● | ● | ● | ● |
| ECOG PS | | ● | | ● | ● | | | | | ● | ● | ● | ● |
| mMRC | | | | ● | ● | | | | | | | | |
| PK/PD | | | | | ● | ● | | ● | | | | | |
| Immune cell assessment | | | | ● | ● | | ● | ● | | ● | | | |
| AEs | | ◄————————————————————————————————————————————————► | | | | | | | | | | | |
| DLTs under hospitalization | | | | ◄——————————————————► | | | | | | | | | |
| HLA test | | ● | | | | | | | | | | | |
| Anti-HLA antibody | | ● | | | ● | | | | | ● | | | ● |

**Fig 1. Study schedule of enrollment, interventions, and assessments.** Abbreviations: ECG: electric cardiogram; CBC: complete blood count; ECOG PS: Eastern Cooperative Oncology Group Performance Status; mMRC: modified Medical Research Council; PK/PD: pharmacokinetic/pharmacodynamic; AE: adverse event; DLT: dose limiting toxicity; HLA: human leukocyte antigen.

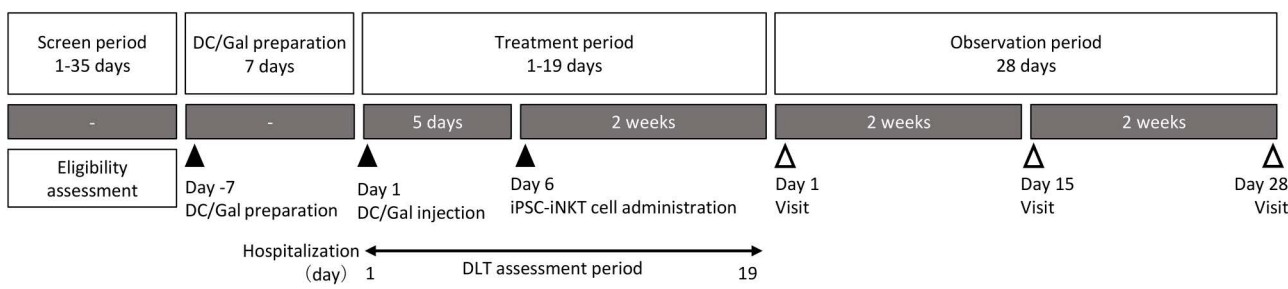

**Fig 2. The study outline, including all study visits and timeline.** Abbreviations: DC/Gal: dendritic cells pulsed with a-Galactosylceramide; DLT: dose limiting toxicity; iPSC-iNKT: induced pluripotent stem cell-derived invariant natural killer T.

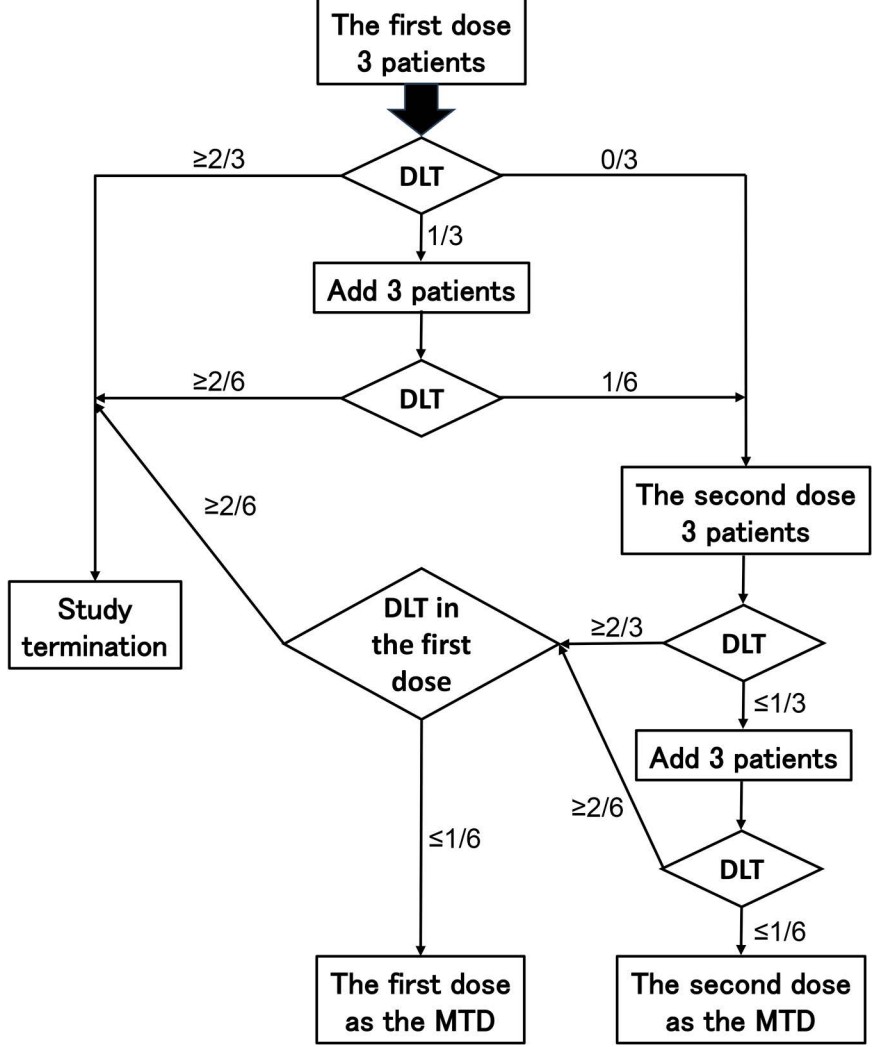

**Fig 3. The 3+3 design for dose escalation.** The descriptions in the rectangles represent the actions needed. The descriptions in the diamonds represent the assessment for DLT. The numbers above the arrays represent the number of patients who have DLT. Abbreviations: DLT: dose limiting toxicity; MTD: maximum tolerable dose.

5) Patients who meet the following laboratory data: hemoglobin ≥ 10 g/dL, white blood cell count ≥ 3000/µL, platelet count ≥ 75,000/µL, serum creatine ≤ 1.5 mg/dL, total bilirubin ≤ 1.5 mg/dL, AST (GOT) and ALT (GPT) < 2.5 times the upper limit of facility reference values, and $SpO_2$ (under room air) ≥ 93%.

6) Patients who are expected to have a prognosis of 3 months or more.

7) Patients for whom written informed consent has been obtained from the individual.

**Exclusion criteria.** Patients who meet any of the following conditions are not eligible.

1) Patients who are Hepatitis B Surface (HBs), Hepatitis C Virus (HCV), Human Immunodeficiency Virus (HIV), or Human T-lymphotropic virus type 1 (HTLV-1) antibody-positive or HBs antibody-negative, but have Hepatitis B Virus (HBV)-DNA detected by HBV-DNA quantitative testing.

2)  Patients who have been taking or injecting corticosteroids (methylprednisolone 10 mg/day or higher or equivalent) or immunosuppressive drugs within at least 2 weeks prior to the start of the study product.

3)  Women who are pregnant, lactating, or planning to become pregnant during the study period and men who do not agree with using any of the effective contraceptive methods under the guidance of a physician during the study period or up to 14 days after the last dose of the study product.

4)  Patients with active autoimmune disease requiring systemic or immunosuppressive therapy with corticosteroids or biologic agents.

5)  Patients who have experienced immune-related AEs with immune checkpoint inhibitor treatment.

6)  Patients with poorly controlled diabetes mellitus.

7)  Patients with severe lung disease (modified Medical Research Council (mMRC) Breathlessness Scale Grade 2 or higher) or with a history of non-infectious interstitial lung disease requiring steroid treatment.

8)  Patients with significant cardiac disease (New York Heart Association (NYHA) class III or greater).

9)  Patients with concurrent multiple cancers.

10) Patients who are unable to use contrast agents in radiography, such as from an allergy or kidney disfunction.

11) Patients with a history of hypersensitivity to human serum albumin products or proteins of foreign origin.

12) Patients who, at the time of informed consent, are participating in other clinical trials or clinical studies and are receiving other investigational products or are judged by the principal investigator (PI) or sub-investigator (SI) to have residual effects of AEs caused by such products.

13) Patients with completely identical genotypes of HLA-A, HLA-B, and HLA-C to the investigational product to distinguish the study product from participants' iNKT cells.

14) Patients who are prohibited from undergoing blood apheresis due to comorbidities, such as unstable angina, A-V block class 2 or greater, Wolff-Parkinson-White (WPW) syndrome, complete left bundle branch block, systolic blood pressure of 90 mmHg or lower, or systolic blood pressure of 170 mmHg or higher.

15) Patients who are judged to be unsuitable to participate in the study.

## Statistical analysis and sample size

The statistical analysis plan for this study is summarized below, with the specific details separately described in the statistical analysis plan. Details of the statistical methods may be further specified in the Statistical Analysis Plan (SAP). If any changes to the primary endpoints or analysis methods are required, they will be implemented through a protocol amendment, and the SAP will be updated accordingly. Patients enrolled in the study and treated with DC/Gal are the target population for safety analysis. However, cases of non-compliance with the Act on the Safety of Regenerative Medicine are excluded from the safety analysis population. Among the safety analysis population, all participants who could not be appropriately evaluated during the DLT evaluation period for reasons other than DC/Gal- or iPSC-iNKT cell-related toxicity are excluded from the DLT evaluation population. If a participant is excluded from the analysis, then a new participant will be added to the cohort.

This study is designed as an exploratory, early-phase evaluation of the safety, tolerability, and exploratory immune changes following treatment with DC/Gal and iPSC-iNKT cell combination therapy. Therefore, no formal hypothesis testing will be performed, and analyses will be descriptive and exploratory in nature.

We employ a 3 + 3 design for this study. The maximum number of patients to be analyzed is set at 12, with six patients in each of the two dose cohorts. The minimum number of cases is set at two, as the study would be terminated because of the occurrence of two consecutive cases of DLT in the first dose cohort.

For each cohort, the number and proportion of patients experiencing a DLT, with corresponding 95% confidence intervals, are calculated. The following secondary endpoints are analyzed for the purpose of providing supplementary discussion to the primary analysis results. For the safety secondary endpoints, frequency tables are created for all AEs occurring after the DC/Gal or iPSC-iNKT cell administration for each system organ class (SOC) and preferred term (PT) in the Medical Dictionary for Regulatory Activities (MedDRA) Japanese version. Similar frequency tables are created by dose, severity, causal relationship to the DC/Gal or iPSC-iNKT cells, and the common terminology criteria for adverse events (CTCAE) grade. Summary statistics are calculated for the laboratory test results and any changes occurring between baseline and each visit. A trend chart of laboratory values are also created. For the efficacy secondary endpoints, the response rate and disease control rate using response evaluation criteria in solid tumors (RECIST) v.1.1 evaluated by the PI or SI are calculated, with the 95% CI for these rates also calculated. This is an exploratory, early-phase study primarily designed to evaluate safety and tolerability. Therefore, no formal hypothesis testing for efficacy is planned and the study is not powered to demonstrate efficacy. Any P values or confidence intervals derived from these analyses will be reported as nominal and exploratory and will not be used to draw definitive conclusions about efficacy. Other items and details are described in the statistical analysis plan.

## iPSC-iNKT cells (investigational product)

The iPSC-iNKT cell product was previously cultured, and stored as a frozen master cell bank using a healthy volunteer who provided written informed consent at the RIKEN institute (Yokohama, Kanagawa, Japan). The PI orders the manufacturing of the investigational product at RIKEN following the eligibility confirmation of participant. The iPSC-iNKT cells are incubated for purification for 10–14 days at RIKEN after the order from the PI, then sent to the study site (Chiba University Hospital) in accordance with the standard operating procedure stipulated separately. The product that has passed the product specification test established in advance and agreed upon with the Pharmaceuticals and Medical Devices Agency (PMDA) will be used for this trial (Table 1). The iPSC-iNKT cells are administered into a tumor feeding artery via super-selective arterial infusion. The dose is $3.0 \times 10^7$ cells/m²/injection for the initial dose cohort or $1.0 \times 10^8$ cells/m²/injection for the second dose cohort.

## Study procedures

After receiving written informed consent from the potential participant, the screening tests are conducted. Participant eligibility is ultimately confirmed by the PI and study coordinating physicians. According to the decision from the coordinating physician, we register the eligible participant to this trial. The study schedule details are described in Fig 1.

**DC/Gal preparation period.** The DC/Gal are manufactured by purifying each eligible patient's peripheral monocytes after apheresis and incubating them with αGalCer for seven days at Chiba University Hospital. The DC/Gal is validated following the product specification tests (Table 2) before injection into the participant.

**Treatment period.** The treatment period begins with the injection of DC/Gal ($1.0 \times 10^8$ cells/injection) into the submucosa of the inferior nasal concha of each participant. The iPSC-iNKT cell product is administered to the tumor feeding artery five days after the DC/Gal injection. Both the DC/Gal and iPSC-iNKT cells are administered only once during this trial. We carefully monitor the participant until two weeks after iPSC-iNKT cell product administration. All the participants must be hospitalized throughout the treatment period.

**Observation period.** All the participants must visit the hospital three times by two weeks (days 1, 15, and 28 of the observation period), with potential AEs carefully monitored during the treatment period. If there are any AEs related to this trial at the final visit or time of discontinuation of this study, we will continue to follow the participant until the AEs relieve.

**Table 1. List of specification tests for the induced pluripotent stem cell-derived invariant natural killer T (iPSC-iNKT) cells.**

**Specification test**

| Test category | Detail | Method | Tentative Evaluation Standard Values |
|---|---|---|---|
| General characteristic | Appearance | Seeing | No adhesion of foreign matter, peeling of labels, etc. No liquid leakage |
| | | | No abnormality in color tone and no foreign matter |
| | Cell morphology | Microscopic observation | T cell-like morphology (floating cell) |
| | Viable cell count (product) | Cell counting | More than $1 \times 10^7$ cells/50mL tube |
| | Viability | Flow cytometry | More than 90% of viability (7AAD-) in the lymphocyte gate |
| | Cell surface marker expression | Flow cytometry | At least 80% of the CD45+cells express CD3, and CD3+cells express NKT-specific TCR (Va24, Vb11) |
| Safety | Cell purity | PCR | Undifferentiated marker (LIN28)-expressing cells negative |
| | Sterility | Culture method | Fungi and common bacteria negative |
| | | Rapid method | Fungi and common bacteria negative (except anaerobic bacteria) |
| | Mycoplasma denial | PCR | Negative |
| | Endotoxin | Colorimetric and turbidimetric methods | Less than the standard value (0.3 EU/mL)[†] |
| Cell function | in vitro IFNγ productivity | ELISA | IFNγ level (Target value: 1.7 times the average of negative samples) |
| | in vitro anti-tumor effect | K562 cell killing potential | Confirmation of anti-tumor effect (Target value: 7%) |

**Process-Related Impurity Testing**

| Safety (impurity) | BSA | ELISA | <50 ng/dose (25 mL) in final product |
|---|---|---|---|
| | Mouse genome | PCR | Below detection limit in final product |

[†]: Endotoxin thresholds reflect the validated limit of detection at the manufacturing site.

**Table 2. List of specification tests for the a-galactosyl ceramide-pulsed autologous dendritic cells (DC/Gal).**

| Test category | Method | Specified value/judgement criteria |
|---|---|---|
| Living cell count | Measurement using a blood cell counting plate | More than $1 \times 10^8$ |
| Cell viability | Measured using a trypan blue stained blood cell calculator | More than 60% |
| Rapid viable bacterial test | Membrane filter method/rapid fluorescent staining | Negative |
| Sterility test | Membrane filter method | Negative |
| Endotoxin test | Gelation method/limit test method | < 0.25 EU/mL[†] |
| Mycoplasma denial test | NAT Method | Negative[‡] |
| Endotoxin test (simplified test) | kinetic colorimetry (Simplified method) | < 1 EU/mL |
| Cell surface antigen assay | Flow cytometry | Of CD45+cells, at least 15% of CD86+cells |
| Impurity test | Macro- and Microscopic observation | No foreign matter |
| Appearance test on product container | Seeing | No abnormalities on the container and package |

[†]: Endotoxin thresholds reflect the validated limit of detection at the manufacturing site;

[‡]: Mycoplasma testing methods are equivalent across products.

## Outcome measurements

**DLTs.** DLTs are defined as dose-limiting adverse events possibly related to the DC/Gal or iPSC-iNKT cell product that occurs during the period referred to as the treatment period. The grade is evaluated using the CTCAE ver. 5.0 Japanese version JCOG edition. The following AEs are included:

1) Grade 4 or higher hematological toxicities;

2) Any non-disease-related blood toxicity requiring any transfusion or granulocyte-colony stimulating factor (G-CSF) administration;

3) Grade 3 or higher non-hematological toxicities (excluding transient clinical laboratory abnormalities, diarrhea, nausea, vomiting, or other manageable systemic symptoms that recovered to grade 2 or below under appropriate treatments); and

4) AEs leading to blood transfusion.

If DLT occurrence is suspected, then the PI can consult the Independent Data Monitoring Committee (IDMC). The IDMC determines if the patient should continue or terminate the treatment.

DLT assessment is evaluated at each step, with the DLTs of the first three participant assessed in the first dose cohort. If any of these three patients experienced the DLT, then we will add an additional three patients. If two or more patients experience a DLT at the first dose cohort, the dose will not be escalated, and the study will be stopped. If one or none of the six patients have DLT, then the dose can be considered to be tolerable. The study can then be continued to assess the second dose cohort in the same manner. Finally, if one or none of the six patients have DLT in the second dose cohort, then the IDMC will define the dose as the MTD. DLT assessment is conducted by evaluating all the safety information, including DLT. The IDMC then decides whether to proceed with the next dose level cohort or the MTD.

**Safety measurement.** We assess the DC/Gal and iPSC-iNKT cell safety information throughout the study. We record all AEs that occurred after DC/Gal or iPSC-iNKT cell administration, regardless of the causality. All the AEs are specified in its terminology using the MedDRA, frequency, and severity.

**Efficacy measurement.** As a preliminarily efficacy measurement of the study treatments, we evaluate the tumors using contrast-enhanced computed tomography (CT) or magnetic resonance imaging (MRI) before the DC/Gal injection and at day 15 of the observation period or the time of discontinuation.

**Exploratory outcomes.** We assess the pharmacokinetics or pharmacodynamics of the iPSC-iNKT cells, as well as the HLA type of the peripheral blood cells to detect the iPSC-iNKT cells in the peripheral blood of the participants. In addition, we conduct an immune cell repertoire analysis and omics analysis of the peripheral blood and tumor specimens to evaluate the effects of the study treatments on the immune system of the participants and tumor microenvironment, respectively. Tumor specimens are optionally collected before the study treatment and/or within two days after iPSC-iNKT cell administration, only if the participant consents to this procedure.

## Ethics consideration and data management plans

This study is conducted in accordance with the Declaration of Helsinki, the Pharmaceutical and Medical Device Act, the Act on the Safety of Regenerative Medicine, and the Japanese GCP for regenerative medicine. This study has been approved by the Certified Review Board for Regenerative Medicine of Chiba University and the regenerative medicine review board of Japanese Ministry of Health, Labour, and Welfare (MHLW), and registered with the Japan Registry of Clinical Trials (jRCT) (jRCT number: jRCTa030220741; URL: https://jrct.mhlw.go.jp/en-latest-detail/jRCTa030220741). Additionally, all the study procedures follow the study protocol and associated standard operating procedures stipulated separately. All participants are well informed and provide written consent to participate in this trial using the informed consent form (S1 File) by the PI before the eligibility screening procedures.

All the study data are collected using the electronic case report forms (eCRFs) and stored through the electronic data capturing system "the Fountayn." Documents or records pertaining to the study are preserved at a storage location deemed appropriate until 30 years have elapsed after the study is discontinued or terminated. These data managements are conducted according to the standard operating procedure of this study stipulated separately.

### Safety considerations

We assess the body weight, vital signs, ECOG PS, tumor status, blood tests, urine test, and radiography for all participants throughout this trial. In addition, the participants need to be hospitalized during the treatment period for intensive observation and prompt identification of DLTs. When the participants are discharged, we ensure that the patient can contact our hospital and nearby medical facilities promptly in case of AEs. We monitor AEs from the time the patient provides consent until the end of the observation period (discontinuation). If AEs with a plausible relationship to the investigational product remain unresolved at the end of the observation period, then the investigations will be continued until the recovery, stabilization, or death from such AEs, whichever occurs earliest. Additionally, the IDMC monitors the safety of the clinical trial and make recommendations regarding the dosage and administration intervals. The IDMC provides recommendations for further safety evaluation at a specified dose level and the determination of the next dose evaluation.

### Trial status

At the time of this submission, patient recruitment has commenced. We started recruitment from March 30, 2023, and will end in December 2027 (six participants recruited at the time of submission), on estimation. All data will be collected by the end of February 2027, and this study will be completed by April 2027. The findings of this trial will be disseminated through peer-reviewed publications and scientific conferences.

### Discussion

This is the first-in-human clinical trial using the combination of DC/Gal and iPSC-iNKT cells for head and neck cancer patients. The iPSC-iNKT cells were established from a healthy volunteer and are stocked as a master cell bank at the RIKEN Institute, while αGalCer/DCs are prepared from participants' peripheral blood in the study site. The combination use of autologous and allogeneic cells requires careful consideration for a feasible study flow. This study will provide not only the tolerability, safety, exploratory efficacy, and biological background of the iPSC-iNKT cells for head and neck cancer, but also important evidence guiding the feasible study design and procedures for the combination use of autologous and allogenic cells. This clinical study is essential for developing the iPSC-iNKT cell therapy toward the next phase 2 study to evaluate its efficacy against malignant tumors. Additionally, this trial protocol can contribute to future clinical trials using separately prepared immune cells products, such as an iPS cell-derived immune cell product and patient-derived adjuvant agents.

As a first-in-human study, we schedule frequent blood draws for the PK/PD evaluation during the first week after the first and second administration of the iPSC-iNKT cells. We use HLA genotyping to detect residual iPSC-iNKT cells in participants' peripheral blood. When doing this, we exclude patients with identical HLA genotypes as the iPSC-iNKT cell product from this study.

We set the administration schedule of the DC/Gal and iPSC-iNKT cells on days 1 and 6 of the treatment period, respectively. This treatment sequence was determined from the previous study using DC/Gal [19]. We showed that the DC/Gal are widely distributed toward the neck lymph area 2–7 days after injection into the inferior nasal concha submucosa [18]. We therefore adopt sequential administration of the DC/Gal and iPSC-iNKT cells to maximize the immunological interaction between them. The starting iPSC-iNKT cell dose ($3.0 \times 10^7$ cells/m$^2$) was determined from a pre-clinical study using severely immunodeficient NOG (NOD/Shi-scid/IL-2Rgamma null) mice (unpublished data). The pre-clinical safety study indicated that the no adverse event level (NOAEL) of the combination use of iPSC-iNKT cells with DC/Gal was $2.0 \times 10^6$

cells/body. This NOAEL is equivalent to $3.0 \times 10^7$ cells/m$^2$ for humans following the FDA "Guidance for Industry Estimating the Maximum Safe Starting Dose in Initial Clinical Trials for Therapeutics in Adult Healthy Volunteers" [22]. From this, we consider this starting dose to be ethically and scientifically acceptable. Furthermore, as iPSC-iNKT cells demonstrated cytotoxic activity against six distinct tumor cell lines, their mechanism of action is expected to be broadly applicable across malignancies. Nonetheless, for this first-in-human trial, enrollment is restricted to patients with head and neck cancer, but in future studies this approach could be applied to other tumors with readily accessible tumor-feeding arteries for targeted administration.

Another notable aspect of the scheduling of this study is the preparation period for iPSC-iNKT cell product and the delays allowed for the treatment initiation. With regenerative medicine, which is associated with long-term storage difficulties, the study products need to be expanded from working cell bank consisting of intermediate cell products after the enrollment of each participant. The iPSC-iNKT cell and DC/Gal preparations take 10–14 days and five days, respectively. We determined the screening duration and treatment initiation time after considering the preparation period for study treatment. We also set the allowance to seven days for the treatment initiation time to account for the delay of cell preparation.

This trial began with only single dose cohort ($3.0 \times 10^7$ cells/m$^2$ of the iPSC-iNKT cells) because the tolerability assessment of the second dose ($1.0 \times 10^8$ cells/m$^2$) of the iPSC-iNKT cells was simultaneously in progress in the previous phase 1 trial at the time this trial was initiated (S5 and S6 Files). After confirmation of the tolerability of the second iPSC-iNKT cell dose, we amended the study design to add the second dose cohort. In addition to the amendment, we added the optional tumor biopsy from the participants to examine the anti-tumor effects and immune landscape of the tumor microenvironment. This tumor biopsy is conducted before and after administration of the combination therapy only for participants with an accessible bulky tumor and who agree with this extra biopsy given the ethical considerations.

This phase 1 study has some limitations. Because the DC/Gal and iPSC-iNKT cells are administered at one time, this trial will not assess the tolerability, safety, or efficacy of repeated dosing, restricting evaluation of long-term or repeated combination therapy. The repetitive use of DC/Gal is not feasible considering the burden of blood apheresis on the patient for preparation. However, the allogenic iPSC-iNKT cell product may be able to be administered repeatedly in future trials. Because more than 99.6% of the iPSC-iNKT product was found to be excreted within 48 hours in the non-clinical study, we believe that the bi-weekly use of this product would not accumulate and increase the risk of use. Another limitation of this study is the participants. We enrolled head and neck cancer patients who were unamenable to standard of care in this study in accordance with "the guideline for Evaluation of anti-cancer agent [23]" and because of its approach for the tumor feeding artery. The study design limits the generalizability of the findings to other malignancies. Nevertheless, the iPSC-iNKT cells showed a killing effect not only in head and neck cancer cells, but also in other cell types, suggesting that the results of this trial would be vital and fundamental for the future application of iPSC-iNKT cells for various cancer patients.

## Supporting information

**S1 File. Example of the informed consent form (translated into English).**
(PDF)

**S2 File. SPIRIT checklist.** Recommended items to address in a clinical trial protocol and related documents.
(PDF)

**S3 File. The latest version (v.3.0) of the full protocol (Japanese original version).**
(PDF)

**S4 File. The latest version (v.3.0) of the full protocol (English version).**
(PDF)

**S5 File. The previous version (v.1.8) of the full protocol (Japanese original version).**
(PDF)

**S6 File. The previous version (v.1.8) of the full protocol (English version).**
(PDF)

## Acknowledgments

We thank Kazue Namiki, Chikako Inamata, and Natsumi Kondo for trial coordinating support. We thank J. Iacona, Ph.D., from Edanz (https://jp.edanz.com/ac) for editing a draft of this manuscript.

## Author contributions

**Conceptualization:** Tomoya Kurokawa, Tomohisa Iinuma, Haruhiko Koseki, Shinichiro Motohashi.

**Data curation:** Yoko Hattori.

**Formal analysis:** Yosuke Inaba.

**Funding acquisition:** Haruhiko Koseki.

**Investigation:** Tomohisa Iinuma, Takahiro Aoki.

**Project administration:** Haruna Ebisu, Tomoha Yanagidaira, Tadami Fujiwara, Tominaga Fukazawa.

**Supervision:** Hideki Hanaoka, Toyoyuki Hanazawa, Shinichiro Motohashi.

**Writing – original draft:** Tomoya Kurokawa.

**Writing – review & editing:** Tomohisa Iinuma, Tadami Fujiwara, Haruhiko Koseki, Toyoyuki Hanazawa, Shinichiro Motohashi.

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
