## [Decision Letter · Decision Letter 0]

27 Oct 2025

Dear Dr. Motohashi,

Thank you for submitting your manuscript to PLOS ONE. After careful consideration, we feel that it has merit but does not fully meet PLOS ONE’s publication criteria as it currently stands. Therefore, we invite you to submit a revised version of the manuscript that addresses the points raised during the review process.

We look forward to receiving your revised manuscript.

Kind regards,

Maria de Fátima Matos Almeida Henriques de Macedo, Ph.D.

Academic Editor

PLOS ONE

Journal Requirements:

H. Koseki and S. Motohashi received grants from BrightPath Biotherapeutics Co., Ltd. All other authors declare that they have no relevant conflicts of interest.

Additional Editor Comments:

Please revise according to the reviewers comments

Reviewers' comments:

Reviewer's Responses to Questions

**Comments to the Author**

1. Does the manuscript provide a valid rationale for the proposed study, with clearly identified and justified research questions?

Reviewer #1: Yes

Reviewer #2: Yes

Reviewer #3: Yes

2. Is the protocol technically sound and planned in a manner that will lead to a meaningful outcome and allow testing the stated hypotheses?

Reviewer #1: Yes

Reviewer #2: Yes

Reviewer #3: Yes

3. Is the methodology feasible and described in sufficient detail to allow the work to be replicable?

Reviewer #1: Yes

Reviewer #2: Yes

Reviewer #3: Yes

4. Have the authors described where all data underlying the findings will be made available when the study is complete?

Reviewer #1: Yes

Reviewer #2: Yes

Reviewer #3: Yes

5. Is the manuscript presented in an intelligible fashion and written in standard English?

Reviewer #1: Yes

Reviewer #2: Yes

Reviewer #3: No

You may also provide optional suggestions and comments to authors that they might find helpful in planning their study.

Reviewer #1: Recommedation

Accept

From a statistical perspective this is very well written protocol.

The protocol indicates (Page 8, lines 135-139) the use of (3 + 3) dose escalating design with two dose levels (Dose 1 and Dose 2. Thus provided the first three patients experience acceptable levels of dose limiting toxicity (DLT) on Dose 1, the next 3 patients will receive Dose 2. This is carefully set out in Figure 3.

The protocol describes carefully (Page 13, lines 244-247) the sensible use of a Confidence Interval at the interpretation stage. I am not sure the P-value will be of much use, but this is a very trivial remark.

Reviewer #2: The submitted protocol adheres to international standards for first-in-human studies and provides insight into testing a promising modern cell therapy. I have only a few minor recommendations.

The introduction should explain how iPSC-iNKTs are an improvement over standard peripheral blood iNKTs and how donor variability is reduced. iPSCs come from a healthy donor, so the text should reference published data showing that standardized preparation consistently produces iPSC-iNKTs with a similar phenotype and function.

It is unnecessary to list the names of commercial cell lines. Mentioning the broad effect and referencing the publication is sufficient.

Within the protocol, I would appreciate a mention that cytokine release syndrome and tumor lysis syndrome are monitored. Although these reactions are not expected, they should be mentioned, along with the markers and how treatment is set up in the case of CRS. Feel free to mention that it is set up according to CAR-T therapy guidelines.

The protocol contains references to tables that do not exist (e.g., Table 20). I believe these are typos in the protocol revisions, but I recommend checking them in future revisions.

Overall, the protocol and the study are of high quality.

Reviewer #3: Kurokawa et al. present a study investigating off-the-shelf iPSC-derived invariant NKT (iPSC- iNKT) cells in combination with α-galactosylceramide-pulsed autologous dendritic cells (DC/Gal). The study’s design is both innovative and timely, given the growing interest in off-the- shelf cellular therapies and the clinical application of iPSC-derived products. Its findings are expected to make a significant contribution to the field of cancer immunotherapy, with broad implications for the development of allogeneic or off-the-shelf cellular therapies, while also advancing the translation of iNKT cell–based therapies.

At the same time, several aspects of the manuscript could be improved to enhance clarity and readability. Attached, I provide detailed suggestions to strengthen the presentation and ensure that the study’s rationale, methodology, and outcomes are communicated clearly and effectively.

**Do you want your identity to be public for this peer review?** For information about this choice, including consent withdrawal, please see our Privacy Policy

Reviewer #1: No

Reviewer #2: No

Reviewer #3: No

---

## [Author Response · Author response to Decision Letter 1]

9 Dec 2025

Journal Requirements:

<Response>

We have again confirmed that our manuscript meets PLOS ONE’s style requirements, as described in the templates.

H. Koseki and S. Motohashi received grants from BrightPath Biotherapeutics Co., Ltd. All other authors declare that they have no relevant conflicts of interest.

<Response>

We revised the Competing Interests statement in the cover letter as follows:

H. Koseki and S. Motohashi received grants from BrightPath Biotherapeutics Co., Ltd. All other authors declare that they have no relevant conflicts of interest. This does not alter our adherence to PLOS ONE policies on sharing data and materials.

<Response>

We will adhere to the data sharing policy on your journal, and already stated as follows in the “Data Availability” section:

No datasets were generated or analyzed during the current study. De-identified research data will be made publicly available upon completion and publication of the study.

<Response>

We added one citation as “[21]” on the previous phase I study using iPSC-iNKT cells which has been published just recently.

Reviewer #1:

From a statistical perspective this is very well written protocol.

The protocol indicates (Page 8, lines 135-139) the use of (3 + 3) dose escalating design with two dose levels (Dose 1 and Dose 2. Thus provided the first three patients experience acceptable levels of dose limiting toxicity (DLT) on Dose 1, the next 3 patients will receive Dose 2. This is carefully set out in Figure 3.

The protocol describes carefully (Page 13, lines 244-247) the sensible use of a Confidence Interval at the interpretation stage. I am not sure the P-value will be of much use, but this is a very trivial remark.

<Response>

We thank the reviewer for the valuable comment regarding the P-value. We agree that the confidence interval provides the primary information for interpreting study results. However, we have included P-values as supplementary information to express the degree of incompatibility between the null hypothesis and the data obtained through our study. We believe that presenting both confidence intervals and P-values will make our results accessible to a broader readership while maintaining statistical rigor.

Reviewer #2:

The submitted protocol adheres to international standards for first-in-human studies and provides insight into testing a promising modern cell therapy. I have only a few minor recommendations.

The introduction should explain how iPSC-iNKTs are an improvement over standard peripheral blood iNKTs and how donor variability is reduced. iPSCs come from a healthy donor, so the text should reference published data showing that standardized preparation consistently produces iPSC-iNKTs with a similar phenotype and function.

<Response>

We recently published an article on the previous phase I study using iPSC-iNKTs. In the paper, the specifications of the iPSC-iNKTs were described. So, we will add some description on the specification of iPSC-iNKTs with reference in the introduction part.

At this time, the data has not been published in a paper. However, the relevant data have been compiled as part of the study, including product quality and nonclinical data, in the protocol (Supporting Information, S4). Therefore, we put the S4 as a reference here.

It is unnecessary to list the names of commercial cell lines. Mentioning the broad effect and referencing the publication is sufficient.

<Response>

As mentioned above, the data have not been published in a paper. We will also revise the main text to eliminate redundant descriptions of cell lines and refer readers to the supporting information describing the in vitro tumor suppressive effects.

Within the protocol, I would appreciate a mention that cytokine release syndrome and tumor lysis syndrome are monitored. Although these reactions are not expected, they should be mentioned, along with the markers and how treatment is set up in the case of CRS. Feel free to mention that it is set up according to CAR-T therapy guidelines.

<Response>

As you mentioned, our monitoring plan includes care for CRS and TLS by vigilant monitoring in the hospital during the first 14 days after administration of study treatment. We will consider including a mention on CRS and TLS in the next protocol amendment.

The protocol contains references to tables that do not exist (e.g., Table 20). I believe these are typos in the protocol revisions, but I recommend checking them in future revisions.

<Response>

Thank you for pointing out the typos. We’ll revise the tables in the next protocol amendment.

Overall, the protocol and the study are of high quality.

<Response>

We appreciate your review and mindful advice.

Reviewer#3

Major Revisions

Lines 129-132:

During the treatment period, we evaluated the dose-limiting toxicity (DLT) from the DC/Gal

injection to 14 days after iPSC-iNKT cell administration under hospitalization. The 28-day

observation period starts 14 days after administration of the iPSC-iNKT cells, including three

visits for the safety evaluation (Fig. 1/2).

As patients are monitored throughout both treatment and peri-treatment phases, it would be helpful to clarify at the outset that the monitoring period consists of two phases: a treatment period, during which patients are hospitalized and closely monitored for dose-limiting toxicities (DLTs), and an observation period, during which patients are followed as outpatients with three visits over 28 days.

<Response>

I agree with your suggestion. It can be more straight forward.

Lines 213-215:

The outline of this study plan may be revised in the statistical analysis plan if the primary

endpoints or the analysis methods are changed.

This sentence may be perceived as not methodologically nor regulatory-compliant as it

stands. I would suggest revising as follows: “Details of the statistical methods may be

further specified in the Statistical Analysis Plan (SAP). If any changes to the primary

endpoints or analysis methods are required, they will be implemented through a protocol

amendment, and the SAP will be updated accordingly”

<Response>

I agree with your suggestion. It can be more accurate.

Lines 223-225:

The purpose of this study is to evaluate the tolerability, safety, and efficacy of the DC/Gal and iPSC-iNKT cell combination therapy. Therefore, a significance test for validation purposes will not be performed.

Efficacy: in the abstract, the authors state that the study aims to “explore the immunological changes that occur following this combination treatment”. Therefore, the term “efficacy” could be reconsidered to better reflect the exploratory objective focused on immunological or immune correlates, rather than efficacy assessment per se. Suitable alternatives may include “immune effects” and “tumor response.”

Significance test: the protocol is meant to describe a non-comparative, exploratory, early-phase study. It is indeed acceptable not to perform formal statistical significance testing. However, I would suggest stating it more clearly and scientifically, for example:

“This study is designed as an exploratory, early-phase evaluation of the safety,

tolerability, and exploratory immune changes following treatment with DC/Gal and iPSCiNKT cell combination therapy. Therefore, no formal hypothesis testing will be performed, and analyses will be descriptive and exploratory in nature.”

<Response>

I agree and replace the language as you suggested.

Lines 244-247:

This paragraph includes language about CIs and hypothesis testing that may sounds in contradiction with the stated non-comparative design of the study. To improve clarity and use neutral language, I would suggest "This is an exploratory, early-phase study primarily designed to evaluate safety and tolerability. Therefore, no formal hypothesis testing for efficacy is planned and the study is not powered to demonstrate efficacy. Any P values or confidence intervals derived from these analyses will be reported as nominal and exploratory and will not be used to draw definitive conclusions about efficacy."

<Response>

I agree and replace the language as you suggested.

Lines 250-255:

Please clarify the manufacturing process. The phrase “was previously manufactured”

implies that no further steps are required beyond product release. In contrast, the

phrases “The PI will order the manufacturing” and “incubated for purification” suggest

that additional procedures are needed and that manufacturing prior to the trial start may not have been completed.

<Response>

As you mentioned, the word “manufactured” here is inappropriate. We will delete the word.

Lines 296-298:

These lines would benefit from clarification. Do the authors mean the following:

“DLTs are defined as dose-limiting adverse events possibly related to the DC/Gal or

iPSC-iNKT cell product that occurs during the period referred to as the treatment period. Other adverse events possibly related to the DC/Gal or iPSC-iNKT cell product, occurring during or outside this period, will be summarized separately.”

<Response>

Adverse events other than DLTs are not separately summarized, but will be included as “all AEs” regardless of causality, which also includes DLTs. Therefore, we would like to use the first sentence of your proposal. The description on “all AEs” is described in the “Safety measurement” section (Lines 322-325).

Line 382:

There are few or no examples

Please clarify whether no or only a few examples have been reported, and provide

references as appropriate.

<Response>

Thank you for pointing out the misleading part. The intention of this part was to highlight the challenges of clinical trials using regenerative medicines, specifically regarding the combination of autologous cells and allogeneic regenerative therapy. It’s also challenging to determine the prevalence of such a type of clinical trial. Therefore, we’ll replace the sentence as follows:

The iPSC-iNKT cells were established from a healthy volunteer and are stocked as a master cell bank at the RIKEN Institute, while αGalCer/DCs are prepared from participants’ peripheral blood in the study site. The combination use of autologous and allogeneic cells requires careful consideration for a feasible study flow.

Lines 389-391:

Additionally, this trial protocol can inform future clinical trials using immunological agents in combination with patient-derived adjuvant agents.

This sentence appears to repeat the information stated in lines 386–387. Immunological (non-cell) agents are in fact under investigation in trials combined with patient-derived adjuvant agents (e.g., NCT03294954).

<Response>

We intended to highlight that this study protocol would provide insights into organizing challenging clinical trials using separately prepared immune cells. Therefore, we’d like to clarify our intention as follow (underlined: amended language):

Additionally, this trial protocol can contribute to future clinical trials using separately prepared immune cell combinations, such as iPS cell-derived immune cell products and patient-derived adjuvant agents.

Lines 411-414:

because iPSC-iNKT cells exhibited cytotoxic activity against six tumor cell lines, we did not restrict the tumor pathology type in this trial. Although only head and neck cancer cases were included, iPSC-iNKT cells could be applied to other malignant tumors with accessible tumor-feeding arteries.

The current phrasing is a bit contradictory: it says the trial did not restrict tumor

pathology type, but then notes that only head and neck cancers were included. Consider rewording this passage, for example: “Furthermore, as iPSC-iNKT cells demonstrated cytotoxic activity against six distinct tumor cell lines, their mechanism of action is expected to be broadly applicable across malignancies. Nonetheless, for this first-inhuman trial, enrollment is restricted to patients with head and neck cancer, but in future studies this approach could be applied to other tumors with readily accessible tumorfeeding arteries for targeted administration.”

<Response>

I agree and replace the language as you suggested.

Lines 418-419:

Regenerative medicine products are challenging to store long-term, so study materials must be expanded from the working cell bank after each participant’s enrollment.

For clarity, it may be helpful to explain what is meant by the bank. For example: “[from the working cell bank] consisting of intermediate cell products [after each participant’s enrollment]”.

<Response>

I agree and add the language as you suggested.

Lines 434-484:

The section begins by stating that there are “several limitations,” but the text that follows largely reads as explanations or justifications rather than clearly stated limitations. To frame the limitations explicitly, they should be presented as constraints of the study design or interpretability, followed by brief rea

---

## [Decision Letter · Decision Letter 1]

9 Jan 2026

Dear Dr. Shinichiro Motohashi,

Thank you for submitting your manuscript to PLOS ONE. After careful consideration, we feel that it has merit but does not fully meet PLOS ONE’s publication criteria as it currently stands. Therefore, we invite you to submit a revised version of the manuscript that addresses the points raised by reviewer 3.

We look forward to receiving your revised manuscript.

Kind regards,

Maria de Fátima Matos Almeida Henriques de Macedo, Ph.D.

Academic Editor

PLOS One

Journal Requirements:

Additional Editor Comments:

Please address comments done by reviewer 3.

Reviewers' comments:

Reviewer's Responses to Questions

**Comments to the Author**

1. Does the manuscript provide a valid rationale for the proposed study, with clearly identified and justified research questions?

Reviewer #1: Yes

Reviewer #2: Yes

Reviewer #3: Yes

2. Is the protocol technically sound and planned in a manner that will lead to a meaningful outcome and allow testing the stated hypotheses?

Reviewer #1: Yes

Reviewer #2: Yes

Reviewer #3: Yes

3. Is the methodology feasible and described in sufficient detail to allow the work to be replicable?

Reviewer #1: Yes

Reviewer #2: Yes

Reviewer #3: Yes

4. Have the authors described where all data underlying the findings will be made available when the study is complete?

Reviewer #1: Yes

Reviewer #2: Yes

Reviewer #3: Yes

5. Is the manuscript presented in an intelligible fashion and written in standard English?

Reviewer #1: Yes

Reviewer #2: Yes

Reviewer #3: Yes

You may also provide optional suggestions and comments to authors that they might find helpful in planning their study.

Reviewer #1: The authors have addressed my very trivial remark about the utility of the P-value in this context. I wish the authors well with this study.

Reviewer #2: As part of the revision process, any shortcomings in the already high-quality work were addressed. I have no further comments regarding the work, and I am looking forward to see the results from clinical trial.

Reviewer #3: COMMENTS TO THE AUTHORS (PONE-D-25-33703R1)

Kurokawa et al. have addressed all the comments and revised their study protocol accordingly.

We acknowledge and appreciate the authors’ efforts to incorporate the suggested revisions. The revised manuscript has improved notably in both clarity for readers and accuracy.

We would suggest three additional final edits to ensure that all sections are sufficiently clear and detailed for readers to appreciate the quality and impact of this important study.

1) Lines 79-81.

Please insert “that” after “responses”. The complete sentence will read as: "They can also promote broader antitumor immune responses that indirectly harm cancer cells by activating NK cells and cytotoxic T lymphocytes (CTLs) through release of interferon (IFN)-γ and other cytokines [12-14]"

2) Tables 1-2.

We thank the authors for providing the rationale for the different methods and thresholds used to assess Mycoplasma and Endotoxin. This information may help readers appreciate the implications of generating and applying two distinct products produced under different conditions and in different facilities. We would suggest including this information in the manuscript as brief table footnotes. For example:

[Table 2 footnote]

"Mycoplasma testing methods are equivalent across products."

[Table 1 and Table 2 footnote]

"Endotoxin thresholds reflect the validated limit of detection at the manufacturing site."

3) Line 441-443.

Please complete the sentence with a main clause. One suitable option is provided below:

"Because the DC/Gal and iPSC-iNKT 442 cells are administered at one time, this trial will not assess the tolerability, safety, or efficacy of repeated dosing, restricting evaluation of long-term or repeated 443 combination therapy."

With these final improvements, we recommend the manuscript for publication.

**Do you want your identity to be public for this peer review?** For information about this choice, including consent withdrawal, please see our Privacy Policy

Reviewer #1: No

Reviewer #2: No

Reviewer #3: No

---

## [Author Response · Author response to Decision Letter 2]

17 Jan 2026

Reviewer#3

Dear reviewer,

We sincerely appreciate your additional comment on our protocol paper. We’ve addressed your comments as below.

We would suggest three additional final edits to ensure that all sections are sufficiently clear and detailed for readers to appreciate the quality and impact of this important study.

1) Lines 79-81.

Please insert “that” after “responses”. The complete sentence will read as: "They can also promote broader antitumor immune responses that indirectly harm cancer cells by activating NK cells and cytotoxic T lymphocytes (CTLs) through release of interferon (IFN)-γ and other cytokines [12-14]"

<Response>

The clarity of this sentence has been enhanced by the contribution of your commentary. We revised it as your advice.

2) Tables 1-2.

We thank the authors for providing the rationale for the different methods and thresholds used to assess Mycoplasma and Endotoxin. This information may help readers appreciate the implications of generating and applying two distinct products produced under different conditions and in different facilities. We would suggest including this information in the manuscript as brief table footnotes. For example:

[Table 2 footnote]

"Mycoplasma testing methods are equivalent across products."

[Table 1 and Table 2 footnote]

"Endotoxin thresholds reflect the validated limit of detection at the manufacturing site."

<Response>

As you kindly advised, these footnotes would be helpful for readers. We revised it to add the footnotes as your advice.

3) Line 441-443.

Please complete the sentence with a main clause. One suitable option is provided below:

"Because the DC/Gal and iPSC-iNKT 442 cells are administered at one time, this trial will not assess the tolerability, safety, or efficacy of repeated dosing, restricting evaluation of long-term or repeated 443 combination therapy."

<Response>

The clarity of this sentence has been enhanced by the contribution of your commentary. We revised it as your advice.

---

## [Editor Report · Decision Letter 2]

21 Jan 2026

A Phase I clinical trial to evaluate the tolerability and safety of an allogeneic iPSC-derived iNKT cell and αGalCer-pulsed autologous DC combination therapy for patients with recurrent and advanced Head and Neck Cancer: A study protocol

PONE-D-25-33703R2

Dear Dr. Shinichiro Motohashi,

We’re pleased to inform you that your manuscript has been judged scientifically suitable for publication and will be formally accepted for publication once it meets all outstanding technical requirements.

Kind regards,

Maria de Fátima Matos Almeida Henriques de Macedo, Ph.D.

Academic Editor

PLOS One
---

## [Editor Report · Acceptance letter]

PONE-D-25-33703R2

PLOS One

Dear Dr. Motohashi,

I'm pleased to inform you that your manuscript has been deemed suitable for publication in PLOS One. Congratulations! Your manuscript is now being handed over to our production team.

Kind regards,

on behalf of

Prof. Maria de Fátima Matos Almeida Henriques de Macedo

Academic Editor

PLOS One